# The Cytogenetic Map of the Nile Crocodile (*Crocodylus niloticus*, Crocodylidae, Reptilia) with Fluorescence In Situ Localization of Major Repetitive DNAs

**DOI:** 10.3390/ijms232113063

**Published:** 2022-10-27

**Authors:** Svetlana A. Romanenko, Dmitry Yu. Prokopov, Anastasia A. Proskuryakova, Guzel I. Davletshina, Alexey E. Tupikin, Fumio Kasai, Malcolm A. Ferguson-Smith, Vladimir A. Trifonov

**Affiliations:** 1Institute of Molecular and Cellular Biology, Russian Academy of Sciences, Siberian Branch, 630090 Novosibirsk, Russia; 2Institute of Cytology and Genetics, Russian Academy of Sciences, Siberian Branch, 630090 Novosibirsk, Russia; 3Institute of Chemical Biology and Fundamental Medicine, Russian Academy of Sciences, Siberian Branch, 630090 Novosibirsk, Russia; 4Japanese Collection of Research Bioresources (JCRB) Cell Bank, Laboratory of Cell Cultures, The National Institute of Biomedical Innovation, Health and Nutrition, Saito-Asagi, Ibaraki 567-0085, Osaka, Japan; 5Department of Veterinary Medicine, University of Cambridge, Cambridge CB3 0ES, UK; 6Department of Natural Science, Novosibirsk State University, 630090 Novosibirsk, Russia

**Keywords:** C-banding, CDAG-banding, FISH, GTG-banding, karyotype, heterochromatin, high-throughput sequencing, repeats, reptile, ribosomal DNA, tandem repeats, telomeric DNA

## Abstract

Tandemly arranged and dispersed repetitive DNA sequences are important structural and functional elements that make up a significant portion of vertebrate genomes. Using high throughput, low coverage whole genome sequencing followed by bioinformatics analysis, we have identified seven major tandem repetitive DNAs and two fragments of LTR retrotransposons in the genome of the Nile crocodile (*Crocodylus niloticus*, 2n = 32). The repeats showed great variability in structure, genomic organization, and chromosomal distribution as revealed by fluorescence in situ hybridization (FISH). We found that centromeric and pericentromeric heterochromatin of *C. niloticus* is composed of previously described in *Crocodylus siamensis* CSI-*Hin*dIII and CSI-*Dra*I repetitive sequence families, a satellite revealed in *Crocodylus porosus*, and additionally contains at least three previously unannotated tandem repeats. Both LTR sequences identified here belong to the ERV1 family of endogenous retroviruses. Each pericentromeric region was characterized by a diverse set of repeats, with the exception of chromosome pair 4, in which we found only one type of satellite. Only a few repeats showed non-centromeric signals in addition to their centromeric localization. Mapping of 18S–28S ribosomal RNA genes and telomeric sequences (TTAGGG)_n_ did not demonstrate any co-localization of these sequences with revealed centromeric and pericentromeric heterochromatic blocks.

## 1. Introduction

Reptiles are a large and paraphyletic class of predominantly terrestrial vertebrates. The taxonomy of the group is complex and has changed repeatedly. Living reptiles comprise turtles, crocodilians, squamates (lizards and snakes), and rhynchocephalians (tuatara). Reptiles show a vast diversity in diploid chromosome number (2n), and karyotype morphology, with various combinations of macro- and microchromosomes, and sex-determination systems [1,2,3]. The genome sequence databases contain only a limited amount of reptile data and the complete analysis of chromosomes and genomes is still lacking for many species.

Crocodilians first appeared 95 million years ago in the Late Cretaceous period, and are the closest living relatives of birds. Together they are combined into a single clade of diapsid reptiles—Archosauria [4,5]. The order Crocodylia is divided into three families (Alligatoridae, Crocodylidae, and Gavialidae), and the number of currently described species ranges from 23 to 27 [6,7,8,9,10,11,12,13]. Cytogenetic data for Crocodylia are usually restricted to the description of chromosome numbers, karyotype composition, and some conventional banding; only a few studies have used molecular cytogenetic tools [14,15,16,17,18,19]. Unlike birds and turtles, crocodilian karyotypes do not contain typical reptile dot-shaped microchromosomes. It has been proposed that all microchromosomes disappeared by fusion events after crocodilian divergence from their ancestors [20].

Repetitive DNA sequences make up a large proportion of every eukaryotic genome, and among these, the majority is satellite DNA, organized in large head-to-tail blocks of tandem repeats [21,22,23]. Satellite DNA can differ in the nucleotide sequence, repeating unit length, and copy number both within the genome and between genomes of different species [24]. Due to the fact that satellite DNA forms long arrays of almost identical repeating units, this part of the genome is still difficult for modern sequencing, assembly, and mapping methods [25,26].

Satellite sequences are a major component of centromeric and pericentromeric heterochromatin (see, for example, [27,28,29,30] for reviews). Despite satellite DNA being earlier considered as “junk”, it has now become clear that these DNA arrays are integral to centromere function and stability and they are functionally significant at the level of the spatial organization of chromosomes, their pairing, and segregation during meiosis [24,30,31,32]. Satellite DNA transcripts are also involved in the formation and maintenance of the heterochromatin structure [33,34]. Very often satellite DNAs are co-localized with other tandemly arranged sequences, such as ribosomal RNA genes (rDNA) and telomeric sequences [35,36].

To date, only a few studies have addressed the characterization of repetitive sequences in crocodilian genomes, and they are mostly focused on such common repeats as telomeric and ribosomal DNA [17,18]. When localizing the telomeric (TTAGGG)_n_ sequences, no interstitial signals were found. Single rDNA sites have been found in the Siamese crocodile (*C. siamensis*, CSI) and all studied Alligatoridae species, except both examined *Paleosuchus* species, in which three chromosome pairs bear these sequences [17,18].

Using high-throughput sequencing, it was found that the frequency and distribution of microsatellites differ between the Indian gharial (*Gavialis gangeticus*), the American alligator (*Alligator mississippiensis*), and saltwater crocodile (*C. porosus*), and their genomic sequences were dominated by dinucleotide repeats [37]. Physical mapping on chromosomes was performed for (CGG)_10_ microsatellites only, which marked from four to eight chromosomes of Alligatoridae species [18]. Analysis of the molecular structure of centromeric heterochromatin in the Siamese crocodile showed the centromeric heterochromatin was composed mainly of two repetitive sequence families—two types of GC-rich CSI-*Hin*dIII family sequences (305 bp and 405 bp) and the 94 bp CSI-*Dra*I [17]. It was proposed that the CSI-*Dra*I-01 fragment is Crocodylidae-specific and present in the *C. niloticus* (CNI) genome, as well as that both types of CSI-*Hin*dIII family sequences are characteristic for all three crocodilian families (the Crocodylidae, Gavialidae, and Alligatoridae) [17]. However, chromosomal localization of both types of the CSI-*Hin*dIII family sequences and the CSI-*Dra*I-01 fragment has not been performed on any other species except for *C. siamensis* [17].

The main aim of this study was the identification, initial description, and cytogenetic mapping of major fractions of repetitive DNAs in *C. niloticus*. Using a combination of low-coverage whole-genome high-throughput sequencing, bioinformatics analysis, and molecular cytogenetics we estimated the genomic abundance of repeat families and physically mapped them on chromosomes by FISH. We tested the distribution of identified satellite sequences in the available reference genomic assemblies of various crocodilian species. In order to improve the Nile crocodile karyotype description, we also performed chromosome mapping of the 18S–28S ribosomal RNA genes and telomeric (TTAGGG)_n_ sequences and made C-banding and chromomycin a3 (CMA_3_)/DAPI staining of the Nile crocodile chromosomes.

## 2. Results

### 2.1. C. niloticus Karyotype

The *C. niloticus* karyotype with 2n = 32 has been described previously [14]. GTG-banding made it possible to unambiguously identify homologous chromosomes in the karyotype of the species (Figure 1). CBG-banding revealed blocks of pericentromeric heterochromatin on all pairs of chromosomes. Less intensely stained heterochromatin blocks were found in the distal regions of chromosome 1 q-arm and interstitial regions of all chromosomes, except for pairs 4, 7, 9, and 15 (Figure 1).

CDAG-staining (Chromomycin A3-DAPI-after G-banding) makes it possible to reveal the AT-/GC-composition and heterochromatin position on differentially stained chromosomes [38]. Intensive CMA_3_-positive pericentromeric blocks were seen in all *C. niloticus* chromosomes except pair 2. Chromosomes 6 and 11 carry large bright CMA_3_-positive blocks in p-arms. Weak 4′,6-diamidino-2-phenylindole (DAPI)-positive blocks were revealed in pericentromeric regions of chromosomes 1 and 2 (Figure 2a). The 18S/28S-rDNA probe gave the only interstitial signal on the pair 11 (Figure 1 and Figure 2b). Telomeric repeat (TTTAGG)_n_ marked the distal parts of all chromosomal arms and did not demonstrate any interstitial signals (Figure 2c).

### 2.2. Bioinformatics Analysis of C. niloticus Repetitive DNA

The genome of *C. niloticus* was sequenced for the first time in this study. BGI MGISEQ-2000 sequencing produced 5,118,575 read pairs (≈1 Gb). After pre-processing, only 3,439,364 read pairs (≈516 Mb) were analyzed with TAREAN for tandem repeats and putative LTR consensus reconstruction. The nine most represented clusters of satellites (Sat) and LTR were used in this work. A list of identified and characterized sequences is given in Table 1.

Seven sequences (CNI-Sat-4, CNI-Sat-19, CNI-Sat-36, CNI-Sat-58, CNI-Sat-67, CNI-Sat-93, and CNI-Sat-96) were identified as satellites. Their genome proportion ranged from 0.046% to 0.59%, and the length of the consensus sequence varied from 31 to 162 bp. GC content varied from 37.65% in CNI-Sat-96 to 65% in CNI-Sat-4. Three sequences (CNI-Sat-4, CNI-Sat-19, and CNI-Sat-67) shared homology with known satDNA previously described for *C. siamensis* [17] (Table 1). CNI-Sat-58 has about 98% identity with the tandem satellite SAT-2_Crp found in *C. porosus.* For CNI-Sat-36 and CNI-Sat-93 no significant homology to previously described repetitive DNAs was found. CNI-Sat-96 has a distant similarity to a fragment of Gypsy-9_OD-I of *Oikopleura dioica*.

The sequences CNI-LTR-48 and CNI-LTR-68 were identified as fragments of putative LTR retrotransposons. Their genome proportion was 0.17% and 0.097%, respectively. CNI-LTR-48 had a 576 bp consensus length and its GC content was 40.97%. The consensus length for CNI-LTR-68 was 5414 bp and GC content was 50.68% (Table 1). Both sequences obviously belong to the ERV1 family of endogenous retroviruses, whereas CNI-LTR-48 represents a reduced fragment with no coding sequences, while CNI-LTR-68 contains all essential ORFs (gag, pol, and env). Different parts of CNI-LTR-68 (from 468 to 2707 bp in size) demonstrate homology to different EVR1 elements from crocodiles and turtles, with the largest percentage of identity with ERV1-2_PSi-I of *Pelodiscus sinensis*.

### 2.3. Comparative Analysis of Satellite Sequences

We carried out a search for similar satellite sequences in the available reference genomic assemblies of various crocodilian species (Table 2). Six sequences (CNI-Sat-4, CNI-Sat-19, CNI-Sat-36, CNI-Sat-58, CNI-Sat-67, and CNI-Sat-96) were revealed in the *C. porosus* genome. CNI-Sat-67 is probably a *Crocodylus*-specific satellite, as it is missing from other families. CNI-Sat-19 has not been identified in the genomes of the Alligatoridae families, but it does occur in Gavialidae. CNI-Sat-93 was specific to the *C. niloticus* genome.

### 2.4. Chromosomal Distribution of Repetitive Sequences

All analyzed repeats, with the exception of CNI-LTR-48, demonstrated a cluster organization and were localized mainly in the pericentromeric regions of chromosomes (Figure 3). The repeats showed specific chromosomal localization.

CNI-Sat-4 was localized in the pericentromeric regions of chromosomes CNI1, 3–16. The signal on CNI16 was very weak (Figure 3a,b). The localization of repeat CNI-Sat-36 was similar to CNI-Sat-4. Although CNI-Sat-36 did not hybridize on CNI1, it labeled the pericentromeric regions of chromosomes 3–16 (Figure 3a,e,f). At the same time, the signal on CNI4 was very weak. The intensity of the signals between the repeats CNI-Sat-4 and CNI-Sat-36 differed greatly.

Localization of CNI-Sat-19 was mainly restricted to pericentromeric regions of CNI chromosomes 1–3, and 16 (Figure 3c,d). CNI-LTR-48 is a dispersed repeat, which forms fairly clear blocks during hybridization in the subtelomeric region of the q-arm of CNI1 and in the pericentromeric regions of CNI6, 9, and 11 (Figure 3e).

CNI-Sat-58 marked the p-arms of chromosomes 13, 14, and 16. At some metaphase plates, it slightly marked the distal part of the p-arm of CNI8 (Figure 3g). CNI-Sat-67 hybridized with pericentromeric regions of all chromosomes, except CNI4. The signal on CNI8 was very weak (Figure 3b,f,h,i).

A low signal intensity was observed during the hybridization of CNI-LTR-68 (Figure 3h). It gave weak signals in the pericentromeric regions of chromosomes 1 and 2, and also weakly marked distal regions of all chromosome arms, similarly to the telomeric repeat.

CNI-Sat-93 hybridized with the pericentromeric regions of chromosomes 5, 7, 10–12, 14, and 15; moreover, it gave an interstitial signal on the CNI9 q-arm (Figure 3g,j). On a pair of chromosomes 12, the signal was observed only on one of the homologs. CNI-Sat-96 marked pericentromeric regions of CNI1-3, 12, and 16 (Figure 3d,i,j). On a pair of chromosomes 12, the signal was observed only on one of the homologs (the same homolog contained CNI-Sat-93).

## 3. Discussion

Tandem repeats content is well investigated in the genomes of only such vertebrates as humans, mice, and some birds [30,39,40,41,42,43,44]. The number of works on other species is steadily increasing (e.g., [45,46,47,48,49,50]). Tandem repeats are highly prevalent at centromeres of both animal and plant genomes, but the repeat monomers vary greatly in composition and sequence length [51]. The size of repeats ranges from a few base pairs (microsatellites) up to several kilobases. Due to the very fast evolution and thus a high propensity for block size polymorphism, repetitive sequences have proven to be good genetic [39,43,52,53] and molecular cytogenetic markers demonstrating size-dependent chromosomal localization in species with clearly fragmented genomes (birds and most reptiles) [54,55], as well as making it possible to distinguish between paralogous chromosomes in the genomes of paleopolyploid species [45,46].

Bird macro- and microchromosomes are very conserved in evolution and often syntenic to turtle macro- and microchromosomes, as shown by comparative mapping of protein-coding genes [56]. Using the example of birds and turtles, it was shown that microchromosomes are enriched in GC-rich genes [57]. Comparative painting studies have demonstrated that rearrangements between macro- and microchromosomes in birds occurred very rarely [58,59]. Homogenization of the centromeric repetitive sequences did not take place between macro- and microchromosomes owing to their structural differences [17]. These structural differences between macro- and microchromosomes are believed to have been highly conserved during the evolution of the Archosauromorpha over 260 million years since this lineage diverged from other diapsids (Lepidosauria) [60,61,62,63]. However, in some birds the phenomenon of repetitive DNA homogenization between macro- and microchromosomes has been detected [64].

Crocodilians lack genome compartmentalization depending on chromosome size, which seems to be a consequence of multiple microchromosome fusions after crocodilian divergence from their common ancestors with birds [20]. Analysis of centromeric repeat sequences in the Siamese crocodile showed that the separation of centromeric repeat sequences into macro- and microchromosome-specific, was lost in Crocodylia, and homogenization of centromeric repeat sequences between all chromosomes, except chromosome 2, took place [17].

Repeated sequences identified here in the Nile crocodile genome (Table 1), showed significant differences in structure, genomic organization, and distribution on chromosomes (Figure 4). Seven sequences (CNI-Sat-4, CNI-Sat-19, CNI-Sat-36, CNI-Sat-67, CNI-Sat-93, CNI-Sat-96, and CNI-LTR-68) were located in the pericentromeric regions of the Nile crocodile chromosomes (Figure 4). In view of the fact that constitutive heterochromatin is composed mainly of repetitive elements, the chromosomal localization of the repeats was perfectly correlated with the location of constitutive heterochromatin blocks revealed by C-banding (Figure 1 and Figure 4).

The GC-rich CNI-Sat-4 repeat with a sequence length of 40 bp (Table 1), showing a high degree of similarity to the CSI-*Hin*dIII family of sequences, was found in the Siamese crocodile genome [17]. These repeated sequences showed a roughly similar localization pattern in the karyotypes of the two crocodilian species. Clone CSI-*Hin*dIII-S01 did not label CSI2, and four small CSI chromosomes, and, similarly, clone CSI-*Hin*dIII-M02 did not label CSI2 but labeled only one of the homologues of one small pair of CSI chromosomes [17], CNI-Sat-4 did not produce a signal on CNI2 and 14 (Figure 4). Interestingly, CSI-*Hin*dIII showed incomplete tandem arrays in the Siamese crocodile, with this family of repeats conserved in three crocodilian families, Crocodylidae, Gavialidae, and Alligatoridae [17]. Our data showing the presence of CNI-Sat-4 in genome assemblies of the three crocodilian families further supports a possible ancestral status of the CSI-*Hin*dIII family of sequences in crocodilians (Table 2).

CNI-Sat-19 and CNI-Sat-67 have a similar consensus length (93 and 94 bp, respectively (Table 1)), and they share a high degree of similarity with the CSI-*Dra*I sequence family identified in the *C. siamensis* genome [17]. However, in this case, the pattern of repeat localization was very different. While the CSI-*Dra*I repeated family was localized in the pericentromeric regions of CSI2 and four small CSI chromosomes [17], CNI-Sat-19 was localized in the pericentromeric regions of CNI1, 2, 3, and 16, and CNI-Sat-67 labeled all CNI pairs except CNI4 (Figure 4). The CSI-*Dra*I repeat has been shown to be organized in long tandem arrays in the Siamese crocodile and categorized as a typical centromeric satellite DNA, with this family of repeats being characteristic only of the genus *Crocodylus* but not of higher-ranking taxa [17]. CNI-Sat-67 may be genus-specific as it is distributed in two *Crocodilus* species, while the detection of CNI-Sat-19 in *Gavialis gangeticus* indicates that the CSI-*Dra*I repeated family is common for at least two crocodilian families (Table 2).

Clusters of three tandem repeats (CNI-Sat-36, CNI-Sat-93, and CNI-Sat-96) found in the Nile crocodile chromosomes bear no homology with the previously described repeat sequences from GenBank, but one of the satellites (CNI-Sat-58) was revealed in *C. porosus* genome by RepBase [65]. Four repeats (CNI-Sat-36, CNI-Sat-93, CNI-Sat-96, and CNI-LTR-68) were localized in the pericentromeric regions of the *C. niloticus* chromosomes, showing chromosome-specific distribution. Like the CSI-*Hin*dIII and CSI-*Dra*I repeat families, there was no obvious attraction of a certain type of repeat to larger or smaller chromosomes in the Nile crocodile karyotype (Figure 4).

Two clusters of repeats (CNI-LTR-48 and CNI-Sat-58) localized outside the centromeric regions (Figure 4). In addition to separate blocks on chromosomes 1, 6, 9, and 11 of the Nile crocodile, CNI-LTR-48 also gave dispersed signals mainly on large chromosomes (Figure 3e). Despite the fact that CNI-Sat-96 gave weak signals in the subtelomeric regions of chromosomes, we did not find homology of this repeat to telomeric sequences.

We noticed that some satellites co-localized on many chromosomes. Thus, CNI-Sat-4, CNI-sat-36, and CNI-Sat-67 are located in pericentromeric regions of most Nile crocodile chromosomes (CNI3, 5–16) (Figure 4). These three satellites together with CNI-Sat-93 were found in the pericentromeric regions of CNI5, 7, 9, and 11–16 (Figure 4). Satellites CNI-Sat-19, CNI-Sat-67, CNI-Sat-96, and CNI-LTR-68 were found in the two largest chromosomes (CNI1 and 2) (Figure 4). This pattern of localization indicates the evolutionary processes driving the linked spreading of satellites across different chromosomes.

Of the sequences identified in this work, CNI-Sat-4, 36, and 93 were GC-rich (Table 1). These repeated sequences were not detected on chromosome pair 2 (Figure 3 and Figure 4), which is in excellent agreement with the results of CDAG-staining showing the enrichment of AT-rich heterochromatin in the pericentromeric region of chromosome 2 (Figure 2a). The GC-rich interstitial heterochromatic block on chromosome 11 corresponds to the ribosomal gene cluster (Figure 2a,b).

CNI-Sat-93 seems to be the only species-specific satellite of *C. niloticus*. All other tandem repeats, except genus-specific CNI-Sat-67, are shared between different crocodilian families. Despite the 80–100 mya radiation [66] no significant divergence of tandemly arranged repetitive elements is revealed, which indicates quite slow rates of molecular evolution in the taxon.

The presence of endogenous retroelements is an expected false-positive result of satellite DNA identification using TAREAN due to the presence of direct terminal repeats [67]. Two LTR retrotransposons discovered here belong to the ERV1 class. It was well established that ERVs represent less than 2% of crocodile genomes and many of the elements are species-specific [68]. The high similarity to ERV elements from the turtle demonstrated here is not surprising, as shared endogenous retroviruses have been found previously in the same study. The dispersed nature of the retrotransposons was confirmed by FISH analysis (Figure 3e,h). Some enrichment in fluorescent signal density was observed in pericentromeric and subtelomeric regions, which indicates an uneven distribution of retroviruses across the *C. niloticus* genome.

## 4. Materials and Methods

### 4.1. Cell Line Establishment and Karyotype Analysis

The *C. niloticus* cells were grown from embryonic tissues obtained from La Ferme aux Crocodiles [16,69] and deposed in the Cambridge Resource Center for Comparative Genomics, Department of Veterinary Medicine, UK. The cell culture was provided to the Institute of Molecular and Cellular Biology, SB RAS, Russia for joint research. The cell line of *C. niloticus* was deposited in the IMCB SB RAS cell bank (“The general collection of cell cultures”, 0310-2016-0002). Chromosome suspensions from the cell culture were obtained in the Laboratory of Animal Cytogenetics, IMCB SB RAS, Novosibirsk, Russia, as described previously [70,71].

CBG-banding was made as described by Gladkikh et al. [72]. GTG-banding was performed prior to FISH using the standard trypsin/Giemsa treatment procedure [73]. CDAG-banding was conducted as described before [38].

### 4.2. DNA Extraction, Library Preparation, and Whole-Genome DNA Sequencing

Whole-genome DNA of *C. niloticus* was extracted from cell culture using Thermo Scientific GeneJET Genomic DNA Purification (Thermo Fisher Scientific Baltics, Vilnius, Lithuania) according to the manufacturer’s protocol. Genomic DNA was fragmented in microTUBE AFA Fiber Pre-Split tubes on a Covaris S2 device (Covaris, Woburn, MA, USA) with a median fragment length of 400 bp according to the manufacturer’s protocol. Libraries were prepared from fragmented DNA using the MGIEasy Universal DNA Library Prep Set (MGI, Shenzhen, China) according to the manufacturer’s protocol. Library sequencing was performed on an MGISEQ-2000 using the DNBSEQ-G400RS High-throughput Sequencing Set (FCL PE100) in the SB RAS Genomics Core Facility (ICBFM SB RAS, Novosibirsk, Russia). The data were deposited in the NCBI SRA database (PRJNA861923).

### 4.3. Repetitive DNA Identification and Characterization

Preprocessing of raw reads was carried out using fastp 0.23.2 [74]. A total of 4,321,634 pairs of high-quality reads with a length of 75 bp were randomly selected and 3,439,364 read pairs (39.1% of genome size [69]) used in the analysis with the TAREAN 2.3.7 tool [67], which identified clusters of the most abundant tandemly arranged repeats. NCBI BLAST [75] and RepBase [65] databases were used to compare consensus tandem repeat sequences with previously described sequences. Putative LTR retrotransposons were analyzed using the Conserved Domain Database [76].

### 4.4. Fluorescence In Situ Hybridization (FISH) with Repeats

Primers for PCR amplification and labeling of nine probes (for repeats CNI-Sat-4, 19, 36, 58, 67, 93, 96, CNI-LTR-48, and CNI-LTR-68) were designed with Gene Runner (version 6.5.52) (Appendix A). In the case of CNI-LTR-48 and CNI-LTR-68, the primers were matched to a region of the sequences of these putative LTR elements. PCR amplification was performed as described earlier [45]. The telomeric DNA probe was generated by PCR with oligonucleotides (TTAGGG)_5_ and (CCCTAA)_5_ [77]. Clones of human ribosomal DNA (rDNA) containing a partial 18S ribosomal gene, the full 5.8S gene, a part of the 28S gene, and two internal transcribed spacers were obtained as described elsewhere [78]. Labeling was performed using PCR by incorporation of biotin-dUTP and digoxigenin-dUTP (Sigma). FISH was performed in accordance with previously published protocols [79]. Images were captured using the VideoTest-FISH software (Imicrotec, New York, NY, USA) with a JenOptic charge-coupled device (CCD) camera mounted on an Olympus BX53 microscope. Hybridization signals were assigned to specific chromosome regions identified by means of GTG-banding patterns photographed by the CCD camera. All images were processed in Adobe PhotoShop 2021 (Adobe, San Jose, CA, USA).

## 5. Conclusions

Despite the important functional role in centromere organization, tandemly arranged repetitive elements remain poorly studied and are mostly missing from genome assemblies. Here we described the major repetitive DNAs of the Nile crocodile (*C. niloticus*) for the first time and mapped the discovered elements to chromosomes. We confirmed four (CNI-Sat-4, CNI-Sat-19, CNI-Sat-58, and CNI-Sat-67) previously described annotated tandemly arranged elements and revealed three (CNI-Sat-36, CNI-Sat-93, and CNI-Sat-96) novel tandem repeats. We demonstrated that almost all major clusters have centromeric and pericentromeric localization, which may assume the role of some of them in centromere assembly. All pericentromeric regions are characterized by a diverse set of repeated sequences, except for chromosome pair 4, in which we found only one type of repeat. Very low rates of molecular evolution of most discovered repetitive elements may suggest their functional significance. The tandemly arranged repeated elements discovered here may be involved in centromere organization, but additional experiments including chromatin anti-CENP-B immunoprecipitation are necessary to resolve this issue. This work complements available genome assemblies and opens new perspectives in repetitive DNA studies of other crocodiles and reptiles in general.

## Figures and Tables

**Figure 1 ijms-23-13063-f001:**
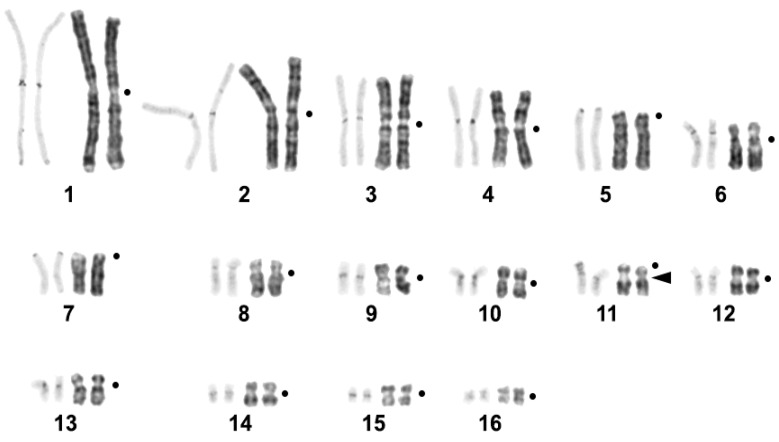
CBG- and GTG-banded *Crocodylus niloticus* karyotype. The black arrow marks the region of 18S/28S-rDNA probe localization. The black dots mark the positions of centromeres.

**Figure 2 ijms-23-13063-f002:**
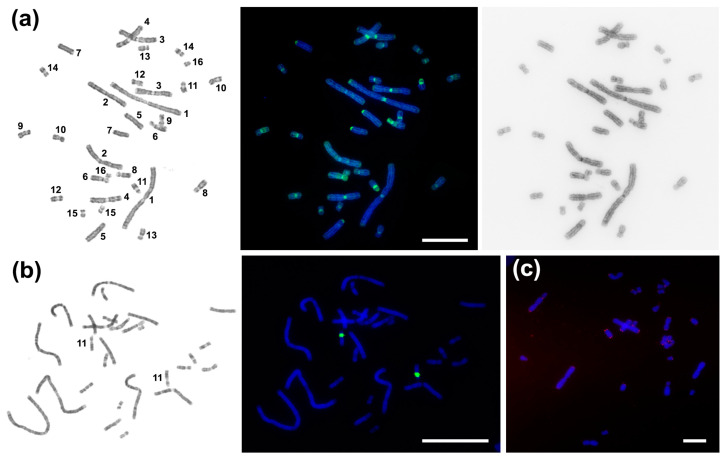
CDAG-staining and localization of 18S/28S-rDNA and telomeric probes on *Crocodylus niloticus* metaphases chromosomes. (**a**) CDAG-staining. From left to right: GTG-banded chromosomes, CDAG-staining, inverted DAPI-banding. Bright blue blocks contain AT-rich DAPI-positive heterochromatin. Bright green blocks consist of GC-rich CMA_3_-positive heterochromatin; (**b**) Localization of 18S/28S-rDNA probe: GTG-banding on the left; (**c**) Localization of telomeric repeat (TTTAGG)_n_. Scale bar 10 µm.

**Figure 3 ijms-23-13063-f003:**
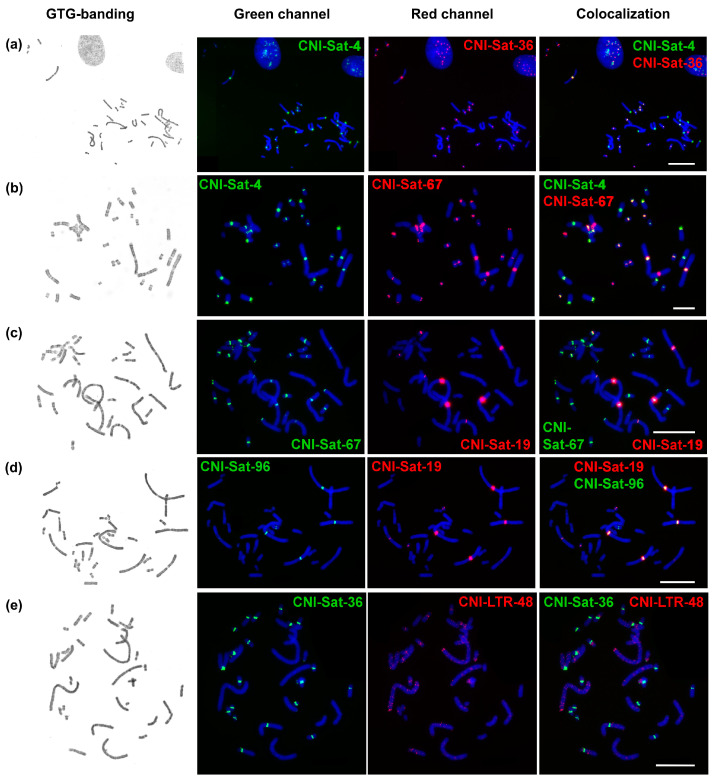
FISH of repetitive DNA probes on metaphase plates of *Crocodylus niloticus*. For each pair of probes from left to right, GTG-banded metaphase, green channel, red channel, and the colocalization of probes are shown: (**a**) CNI-Sat-4 (green) and CNI-Sat-36 (red); (**b**) CNI-Sat-4 (green) and CNI-Sat-67 (red); (**c**) CNI-Sat-19 (red) and CNI-Sat-67 (green); (**d**) CNI-Sat-19 (red) and CNI-Sat-96 (green); (**e**) CNI-Sat-39 (green) and CNI-LTR-48 (red); (**f**) CNI-Sat-36 (green) and CNI-Sat-67 (red); (**g**) CNI-Sat-58 (green) and CNI-Sat-93 (red); (**h**) CNI-Sat-67 (red) and CNI-LTR-68 (green); (**i**) CNI-Sat-67 (green) and CNI-Sat-96 (red); (**j**) CNI-Sat-93 (red) and CNI-Sat-96 (green). Scale bar 10 µm.

**Figure 4 ijms-23-13063-f004:**
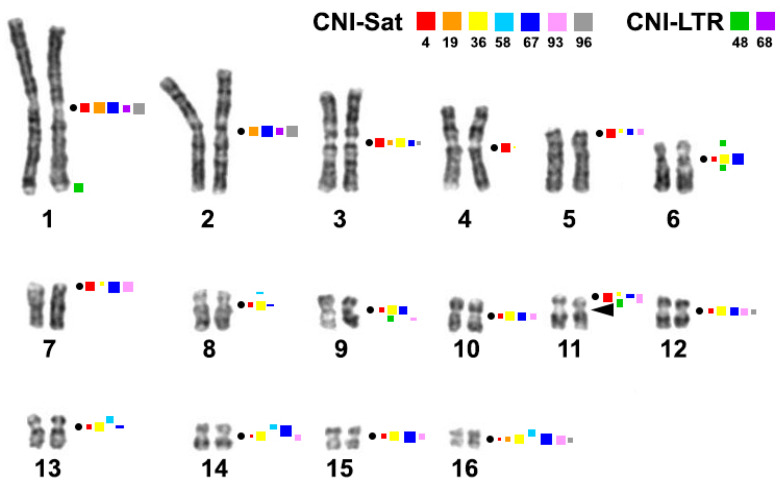
Repetitive DNA probe location on the GTG-banded *Crocodylus niloticus* karyotype. Each cluster of repeated sequences has its own color code (shown in the upper figure right corner). The sizes of the squares roughly show signal intensity. The black arrow marks the region of 18S/28S-rDNA probe localization. The black dots mark the positions of centromeres.

**Table 1 ijms-23-13063-t001:** Putative satellites (Sat) and putative LTR elements (LTR) revealed in the *Crocodylus niloticus* genome.

Repetitive Sequence Name	Genome Proportion	Consensus Length	GC-Content	Accession Number	Homology to Known DNA (Best Hit)	Query Coverage	Percent Identity
CNI-Sat-4	0.59%	40	65%	OP480173	*Crocodylus siamensis* DNA, centromere-specific GC-rich repetitive sequences: clones CSI-*Hin*dIII-M02, CSI-*Hin*dIII-S40, CSI-*Hin*dIII-S07 (GenBank AB434504.1; AB434503.1; AB434500.1)	100%	97.5%
CNI-Sat-19	0.3%	93	40.86%	OP480174	*Crocodylus siamensis* DNA, centromere-specific repetitive sequences: clones CSI-*Dra*I-01, CSI-*Dra*I-05 (GenBank AB434505.1; AB434506.1)	95%	85.56%
CNI-Sat-36	0.24%	101	60.4%	OP480175	-		
CNI-Sat-58	0.12%	112	39.29%	OP480176	*Crocodylus porosus,* satellite repeat, SAT-2_Crp		~98%
CNI-Sat-67	0.1%	94	44.68%	OP480177	*Crocodylus siamensis* DNA, centromere-specific repetitive sequences: clone CSI-*Dra*I-05 (GenBank AB434506.1)	79%	90.67%
CNI-Sat-93	0.05%	31	67.74%	OP480178	-	-	-
CNI-Sat-96	0.046%	162	37.65%	OP480179	*Oikopleura dioica*, part of Gypsy-9_OD-I	-	71.7%
CNI-LTR-48	0.17%	576	40.97%	OP480180	*Crocodylus porosus*, ERV1-2B_Crp-LTR	-	85.64%
CNI-LTR-68	0.097%	5414	50.68%	OP480181	*Pelodiscus sinensis*, ERV1-2_PSi-I	-	80.68%

**Table 2 ijms-23-13063-t002:** Representation of satellite sequences in genome assemblies of four Crocodylia species. Percentage of identity for the best hit (obtained using blastn) shown for each satellite.

Satellites Revealed in *Crocodylus niloticus* Genome	*Alligator sinensis* (GCA_000455745.1)	*Alligator mississippiensis* (GCA_000281125.4)	*Gavialis gangeticus* (GCA_001723915.1)	*Crocodylus porosus* (GCA_000768395.2)
CNI-Sat-4	84.62%	89.47%	87.18%	100%
CNI-Sat-19	-	-	74.68%	94.62%
CNI-Sat-36	81.82%	78.08%	79.73%	96.04%
CNI-Sat-58	83.81%	84.26%	88.89%	97.27%
CNI-Sat-67	-	-	-	89.77%
CNI-Sat-93	-	-	-	-
CNI-Sat-96	73.38%	72.66%	77.03%	82.53%

## Data Availability

The sequenced genomic reads were deposited in the NCBI SRA database under the accession number PRJNA861923. The obtained repetitive element consensus sequences were deposited in the GenBank database (OP480173-OP480181).

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
