# Peer review of "The Cytogenetic Map of the Nile Crocodile (Crocodylus niloticus, Crocodylidae, Reptilia) with Fluorescence In Situ Localization of Major Repetitive DNAs"

_ijms, 2022, doi:10.3390/ijms232113063_

Round 1

Reviewer 1 Report

The paper from Romanenko and coworkers on cytogenegtic map of nile crocodile is a very interesting work which provides new relevant information on cytogentic composition of crocodiles that has been poorly studied till now. I suggest to accept the paper for publication in its presen form.

Author Response

We thank the reviewer for the high evaluation of our work.

Reviewer 2 Report

The manuscript by Romanenko et al. describes the localization (by FISH) of major repetitive sequences (9 were selected) found in the genome of the Nile crocodile. Although it is a descriptive work, a detailed analysis/description of all repetitive sequences identified in CNI, and not only those with enough representation to be detected by FISH (“major fraction”), was expected.

Other major issues I find with the manuscript are:

-     -     As stated before, deeper analysis of repetitive sequences would be expected for a manuscript sent to a SI devoted to high-throughput sequencing in genomics (HSG). Also, taking into account that there are genome assemblies available for 4 Crocodylia species (A. sinensis, A. mississippiensis, G. gangeticus and C. porosus), a comparative analysis of the repetitive sequences in all these species would also be expected.

-      -    It would also be possible to BLAST the satellite DNAs found in CNI in the genomes of the other Crocodylia species.

-       -   Regarding sentence “Each of pericentromeric regions was characterized by a diverse set of repetitive sequences, with the exception of chromosome pair 4, in which we found only one type of repeat, indicating an important role of repetitive sequences in the evolution of crocodilian genomes.” What is the relationship between the location (centromeric) of the analysed repetitive sequences in C. niloticus and their role in the evolution of crocodilian genomes? The same applies to sentence in L250-252: “Different patterns of chromosomal localization point to the important role of repeated sequences in the evolution of crocodilian genomes and some may be involved in functional crocodylian centromere assembly”.

-        -  How is it possible to talk about repetitive homogenization without comparing centromeric sequences from different chromosomes? On the other hand, the absence of FISH signal does not mean the repetitive sequence is not located in a particular region. 

Minor comments:

-       -   Keywords: ribosome and telomere – ribosomal and telomeric DNA

-        -  Please, indicate the meaning of CSI the first time the abbreviation arises (abstract). Unfamiliar readers understand the meaning by the middle of the manuscript, when the name of the other species is mentioned (L72).

-        -  Use the same format for the number of reads in material and method (e.g. 8643269) and result sections (e.g. 8,643,269).

-        -  Use letters to refer to specific pictures in Figure 4.

-         - L221: CSI-DraI instead of CSI-dry repeat?

-         - The telomeric signal is not visible in the figure (Fig. 3C).

-         - I have problems with the numbers:

A) L279: 6878728 are reads, or pair of reads? Taking into account they are 39,1% of genome size I suppose the amount refers to reads. Please, clarify in the manuscript. Also, in results it is mentioned 3439364 read pairs. I would always use the same units.

B) If 5118575 read pairs were produced, how is it possible to select 8643268 pairs? L125: “BGI MGISEQ-2000 sequencing produced 5,118,575 read pairs (≈1 Gb)”

L278: “A total of 8643268 278 pairs of high-quality reads with a length of 75 bp were randomly selected”

-       -   L309: use V.A.T instead of VAT (as before)

Author Response

We sincerely thank the reviewer for all comments, the work on which allowed us to improve our article. We upload the file with responses for the reviewer’s comments.

Reviewer 3 Report

In this manuscript, Authors describe the karyotype of a crocodile species using chromosome banding and staining techniques and fluorescent in situ hybridization of highly repetitive probes. I consider that the integration of both technologies (FISH + banding) provides a greater degree of credibility to the cormosomal maps generated. The methods are adequately explained, the text is perfectly legible (although there are some typos here and there. i.e. "subtelomericl", L154)  and the quality of the photographs and hybridizations is high for this kind of studies. 

I would ask the authors to spend a few lines about the syntenic association of these satellites at the centromeres. In this sense, I know that repeatexplorer provides information about the interconnectivity of NGS reads between satellite families and perhaps they could further demonstrate collocation by this means.

Please include size bars and their equivalence on your images.

Author Response

(The authors gave the same response as above.)

Reviewer 4 Report

-General comments:

The ms entitle “The cytogenetic map of the Nile crocodile (Crocodylus niloticus, Crocodylidae, Reptilia) with in situ localization of major repetitive DNAs” is original because it sequenced the Nile crocodile genome for the first time. After, repeated sequences of interest were localised by FISH in the chromosomes. The authors have done a very thorough job applying various methodologies and obtaining sufficient results of interest for publication. However, the manuscript is hard to read as it is very disorganised with incomplete information, which makes it lose quality, thetefore should be rewritten.

-Specific commnets:

Title: is poor, it should be more informative and attractive.

Change “in situ” to “fluorescence in situ hybridization”

 Abstract: rewrite it

-Should be include the number of the chromosomes of the Nile crocodile.

-Order the abstract according to the order of the results, with the most relevant conclusion at the end of the abstract.

-Line 22: define CSI.

-Line 25: include the types of teh repeat sequences

 Introduction section:

-Lines 65 -67: the sentence is not understood, rewrite it.

-Wrtite the objectives of the work clearly and concisely, in order and accordance with the results. Please, make it clear if this is the first time the crocodile (Crocodylus niloticus) genome has been sequenced.

 Results: Rewrite it

In general this section is unorganised, which makes it difficult to understand. Moreover, it lacks information. Part of the results are descibed in the discusión section.

I suggest as logical order: (1) description of karyotipe, (2) Bioinformatic analysis, and (3) all FISH results

 -Sub-section 2.1. C niloticus karyotype: This specie has 2n = 32. Also, the morphology of the chromosomes should be described and the criteria for the metacentric chromosomes (for expample, is q arm, the down arm?). Define which is the NF (Number Fundamental). In this sub-section the authors also could include GtG-banded and CBG banded resuls (without results of the FISH) and explain them.

-Figure 1 is not understood and should be includes in the last sub-section of the results, as resume of the all FISH results. In the caption there are part of the results (lines 107-110).

-Figure 2: lines 116 – 119 are results; include in the text.

 Sub-section: Bioinformatic analyis: It should start with an introductory sentence, e.g. "the genome of C niloticus was sequenced for the first time in this study..." (If it has been sequenced for the first time, it would be something novel to highlight, and should be include also this point clearly in the objetives of this work). Pherhaps, sequencing data should be includes in a table in supplementary materials and comment and referenced the new table in the text.

-Explain the results of the Table 1.

 Sub-section of the distribution of repetitive sequences. Include all Figures of the FISH, Figure 3, Figure 4 and finally, Figure 1. Describe the results and the Tables in order, clearly and concisely.

 -Figure 3c is of poor quality and the signals of the hybridization are not clear.

- In all figures of FISH, the scale bar of the chromosomes should be included.

-Please, define “sat” the first time that is named

-Figure 4a is of poor quality and chromosomes are not distinguishable

-Lines 147 – 166: expalin the results referencing Figure 4 (a – j)

Discusion: Rewrite it

Lines 197 – 252: In these paragraphs there are a lot of results that the authors could include in the results section.

A conclusión paragraph should be included to collect the most novelty and relevant results and the impact for the advancement of research on the species (C. niloticus).

References section: Check the nomenclature of the species, tehy have to be in italics.

Author Response

(The authors gave the same response as above.)

Round 2

Reviewer 2 Report

The authors corrected all minor comments as suggested. However, they did not answer satisfactorily any of the major concerns indicated.

Author Response

We performed an additional bioinformatic analysis of all revealed sequences and added these results in the article.

Reviewer 4 Report

Comments to the authors:

In this revised versión, the authors have improved the ms, but some aspects related to figure 1 should be modified.

Title: Why don´t you write “FISH” instead “fluorescence in situ”?

Abstract

-The FISH technique, which has been applied to locate the repeated sequences, should be included in the Abstract.

-Line 22: change “C. siamensis” to “Crocodylus siamensis” (is the first time that it is written).

Introduction

-In line 90: change “Crocodylus siamensis” to “C. siamensis”.

-Line 372: Change “Crocodylus niloticus” to “C. niloticus

Results

-Figure 1: I understand the authors with respect to reduce the number of figures in the ms, and this figure is effectively named after mentioning the GTG-banding. But, the location of the signals of the repeat sequences (CNI-sat 4, 19, 36, 58, 67, 93, 96, and CNI-LTR 48, 68) which are shown on GTG-banding, are results obtained after carried out the FISH and should be changed to 2.3 section of the results.

In the 2.1 section of the results, only the figures of the GTG-banding (without the scheme of the location of repetitive sequences) and the CBG-banding should be included, and change the legend of the figure 1.

In section 2.3 of the results, the data of the locations of the repeated sequences should be included after the FISH figures, or even in the discussion section. You could include an ideogram where you represent the scheme of the signals as in figure 1 (CNI-sat, 4, 19, etc. and CNI-LTR 48, 68).

If you include this signals in the section 2.1 of the results (as it is currently), at least you should describe them and explain why are included here.

-On the other hand, in relation to the description of the karyotype (section 2.1 of the results), I agree with the authors in their comments, but the morphology of the chromosomes should be described, at least how many are metacentric, sub-metacentric or telocentric.

Legends figures and tables: the specie should be written with the complete scientific name. Change “C. niloticus” to “Crocodylus niloticus”

-Line 177: The legend of the figure 4 is conffuse. Please, revise

- Please, emphasise the novelty of the work, both in the objectives and in the conclusions.

Author Response

Please find the file attached

Round 3

Reviewer 4 Report

This version of the ms is much improved, clear and easy to follow. I only suggest a few minor changes.

Abstract:

-In lines 21-22 include “FISH” in parenthesis after the definition.

Results:
-Line 135: define CMA3

-Figure 2: write more clear the legend, it is not understood

-Introduce in the text he CDAG-staining significance and why it is done

 -Line 185: change Crocodylus porosus to C. porosus

-Table 2: change C. niloticus to Crocodylus niloticus

 -Figure 4: you can include this Figure in discussion section

Discussion:

Unify the nomenclature of species. In the results it is written for example C. siamensis (scientific name) and in the discussion Siamese crocodile (common name). Please, choose a nomenclature and review the entire ms. I recomend the authors the scientific name.

Conclusion:

-Line 486: include the four tandemly arranged elements, and the three novel tándem repeats to which the text refers.

 -Line 496: remove “some” and include these repeated elements discovered which may be involved in centromere organization

Author Response

We are very grateful to you for the careful reading of the text. Undoubtedly, the work on the remarks made it possible to significantly improve the article.

This time, all comments were also taken into account and corrections were made to the text. The only thing we decided not to unify the nomenclature of species. Since the requirements of the journal do not specify the need to use only scientific name or common, we decided not to make corrections to the text. The practice of using both common and scientific names in articles is widespread in journals of various levels, for example, see https://www.nature.com/articles/s41437-022-00558-6